# The Prognostic Value of the New Combined Hemo-Eosinophil Inflammation Index (HEI Index): A Multicenter Analysis of Anal Cancer Patients Treated with Concurrent Chemo-Radiation

**DOI:** 10.3390/cancers13040671

**Published:** 2021-02-07

**Authors:** Margherita Rimini, Pierfrancesco Franco, Berardino De Bari, Maria Giulia Zampino, Stefano Vagge, Giovanni Luca Frassinetti, Francesca Arcadipane, Almalina Bacigalupo, Martina Valgiusti, Deborah Aloi, Lorenzo Gervaso, Renzo Corvò, Giulia Bartolini, Marianna Alessandra Gerardi, Stefano Cascinu, Andrea Casadei-Gardini

**Affiliations:** 1Department of Oncology and Hematology, Division of Oncology, University Hospital Modena, 41000 Modena, Italy; 73828@studenti.unimore.it; 2Department of Oncology, Radiation Oncology, University of Turin, 10126 Turin, Italy; farcadipane@cittadellasalute.to.it; 3Department of Radiation Oncology, University Hospital of Besançon, 25000 Besançon, France; berardino.debari@rhne.ch (B.D.B.); aloi.deb@gmail.com (D.A.); 4Department of Radiation Oncology, Réseau Hospitalier Neuchâtelois, 2300 La Chaux-de-Fonds, Switzerland; 5Division of Gastrointestinal Oncology and Neuroendocrine Tumors, European Institute of Oncology, IRCCS, 20019 Milan, Italy; maria.zampino@ieo.it (M.G.Z.); lorenzo.gervaso@ieo.it (L.G.); 6Department of Radiation Oncology, IRCCS Ospedale Policlinico San Martino, 16121 Genova, Italy; stefano.vagge@unige.it (S.V.); almalina.bacigalupo@hsanmartino.it (A.B.); 7Department of Medical Oncology, IRCCS Istituto Romagnolo per lo Studio dei Tumori ‘Dino Amadori’–IRST, 47014 Meldola, Italy; luca.frassinetti@irst.emr.it (G.L.F.); martina.valgiusti@irst.emr.it (M.V.); giulia.bartolini@irst.emr.it (G.B.); 8Department of Radiation Oncology, IRCCS Ospedale Policlinico San Martino and Health Science Department (DISSAL), University of Genova, 16121 Genova, Italy; renzo.corvo@unige.it; 9Division of Radiotherapy, European Institute of Oncology, IRCCS, 20019 Milan, Italy; marianna.gerardi@ieo.it; 10Department of Oncology, IRCCS San Raffaele Scientific Institute Hospital, 20019 Milan, Italy; cascinu@yahoo.com (S.C.); casadeigardini.andre@hsr.it (A.C.-G.)

**Keywords:** squamous cell anal cancer, prognostic factors, inflammation, prognostic index

## Abstract

**Simple Summary:**

Bio-humoral predictors of response to chemo-radiation in anal cancer patients are limited, but they might be a useful tool to select patients according to the risk of recurrence, to offer personalized treatments and follow-up protocols during and after therapy. We previously demonstrated the prognostic value of baseline Systemic Inflammation Index (SII), baseline hemoglobin level, and eosinophil count in this clinical setting. In the present study, we proposed a simple scoring system that includes all these three parameters, and we assessed its prognostic value. If validated, this new scoring system could represent a simple tool that is able to drive clinical decision-making in anal cancer patients treated with chemo-radiation with definitive intent.

**Abstract:**

Anal squamous cell carcinoma (SCC) is a rare tumor, and bio-humoral predictors of response to chemo-radiation (CT-RT) are lacking. We developed a prognostic score system based on laboratory inflammation parameters. We investigated the correlation between baseline clinical and laboratory variables and disease-free (DFS) and overall (OS) survival in anal SCC patients treated with CT-RT in five institutions. The bio-humoral parameters of significance were included in a new scoring system, which was tested with other significant variables in a Cox’s proportional hazard model. A total of 308 patients was included. We devised a prognostic model by combining baseline hemoglobin level, SII, and eosinophil count: the Hemo-Eosinophils Inflammation (HEI) Index. We stratified patients according to the HEI index into low- and high-risk groups. Median DFS for low-risk patients was not reached, and it was found to be 79.5 months for high-risk cases (Hazard Ratio 3.22; 95% CI: 2.04–5.10; *p* < 0.0001). Following adjustment for clinical covariates found significant at univariate analysis, multivariate analysis confirmed the HEI index as an independent prognostic factor for DFS and OS. The HEI index was shown to be a prognostic parameter for DFS and OS in anal cancer patients treated with CT-RT. An external validation of the HEI index is mandatory for its use in clinical practice.

## 1. Introduction

Squamous cell carcinoma (SCC) of the anal canal is considered a rare tumor, since it accounts for only 1.5% of all gastrointestinal malignancies [1]. Radiotherapy (RT) in combination with 5-fluorouracil (5-FU) and mitomycin is the standard of treatment for localized disease, resulting in loco-regional recurrence rates ranging from 20% to 40% and overall survival (OS) rates at 3–5 years ranging from 65% to 78% [2,3,4,5,6]. Due to its rarity, only a few prognostic factors has been recognized in anal cancer. From the ACT-I trial and a confirmatory retrospective analysis, palpable lymph nodes and male gender resulted to be potential clinical prognostic factors for loco-regional recurrence and OS [7]. Moreover, a low baseline hemoglobin (Hb) level has been shown to predict for cancer-related death and death for any cause in the same setting [7,8]. The positivity to Human Papilloma Virus infection was previously highlighted to predict for OS and disease-specific survival [9]. The results of the RTOG 98–11 trial showed a significant correlation between male gender and nodal involvement and loco-regional recurrence, as well as between tumor size (>5 cm) and disease-free survival (DFS) and OS [10]. Finally, the European Organization for Research and Treatment of Cancer trial 22861 (EORTC 22861) reported that skin ulceration, nodal involvement, and male gender were prognosticators for survival and local control [3].

However, no laboratory index is currently validated in order to predict the prognosis and guiding clinical decision-making in this setting. In the last few years, research interest has grown on the interplay between cancer, inflammation, and the immune system and several bio-humoral immune-based prognostic scores, such as Lymphocyte count, Neutrophil–Lymphocyte ratio (NLR), and Platelet–Lymphocyte ratio (PLR), have been identified as predictors of survival, recurrence, and treatment response in cancer patients [11,12,13]. In patients affected with anal cancer and treated with concurrent chemo-radiation (CT-RT), our group highlighted the prognostic value of the Systemic Index of Inflammation (SII) [14], low baseline Hb level [8], and high baseline eosinophil level [15] in predicting for poor response to CT-RT.

In this study, we developed a simple scoring system based on the laboratory inflammation values investigated in our previous studies, thus creating a prognostic tool to be potentially used in the decision-making process during clinical practice.

## 2. Materials and Methods

### 2.1. Patient Selection

Clinical data were collected into electronic data files by each participating center co-investigator and double-checked at the data management center. Written informed consent for treatment was obtained from all patients. The study was conducted in accordance with the Declaration of Helsinki, and the protocol was approved by the Ethics Committee of AOU Citta’ della Salute e della Scienza, Turin, Italy (Project identification code: 0089578). The trial was registered in the internal repository for clinical trials at AOU Citta’ della Salute e della Scienza, Turin, Italy (Project identification code: 1190/2016). Patients were treated from March 2007 to February 2019 at the Radiation and Medical Oncology Departments of five institutions (University of Turin, AOU Citta’ della Salute e della Scienza in Turin, Department of Modena Cancer Center, Università di Modena e Reggio Emilia, San Martino Hospital in Genova, Centre Hospitalier Régional Universitaire “Jean Minjoz”, Besançon and European Institute of Oncology in Milan), and clinical data were retrospectively collected. Briefly, all patients had a histological diagnosis of squamous cell carcinoma of the anus obtained via endoscopic examination and were staged with pelvic magnetic resonance, chest computed tomography, and whole body ^18^F-fluoro-2-deoxy-D-glucose positron emission tomography (^18^FDG-PET), according to the 7th Edition of the American Joint Committee on Cancer staging manual [16]. Patients with clinical stage T1–T4, N0–N3, and M0 were included. Those having clinical T1 N0 tumors of the anal margin were excluded when treated with local excision. All patients underwent definitive CT-RT. Chemotherapy consisted of 5-fluorouracil (5-FU) (1000 mg/m^2^/day) given as continuous infusion for 96 h (days 1–5 and 29–33) combined with mitomycin C (MMC) (10 mg/m^2^) given as bolus (days 1 and 29) [17]. Mitomycin C was capped at 20 mg maximum. At the European Institute of Oncology in Milan, 69 patients (22% of the entire population) were treated with CT consisting of 5-FU 200 mg/m2 given as continuous infusion for 24 h combined with Cisplatin 80 mg/m2 (days 1 and 21) or Capecitabine 825 mg/m^2^ twice daily for 5 days/week combined with Cisplatin 60–70 mg/m^2^ every 3 weeks. Radiation was delivered using static or volumetric intensity modulated radiotherapy up to a total prescription dose to the macroscopic disease of 50.4 Gy (48 patients; 15%), 54 Gy (215 patients; 70%), or 59.4 Gy (45 patients; 15%) in 28, 30, or 33 fractions respectively, depending on tumor size and stage. Elective nodal irradiation was offered to patients on pelvic lymph nodes and inguinal groins up to a conventionally fractionated dose of 45 Gy in 30 fractions. The inguinal region was comprised within elective treatment volumes in 285 patients (92%). For patients receiving a simultaneous integrated boost (SIB), 1.8 Gy were given daily to the macroscopic disease and 1.55 Gy were daily administered to the elective volumes [18]. The caudal limit of target volumes was at the anal verge or areas of peri-anal skin involvement. The most cephalad aspect was set where the common iliac vessels bifurcate into external/internal iliacs. The mean relative overlap volume between treatment volumes and pelvic bones was 12.2% (standard deviation: ±5.2%). Response to treatment was assessed, using the Response Evaluation Criteria in Solid Tumours (RECIST), at 6 weeks after treatment and thereafter at 3 and 5 months after CT-RT.

Acute toxicities were scored according to the Common Toxicity Criteria for Adverse Events scale v4.0 (CTCAE v4.0) [19]. In particular, hematological (neutropenia, thrombocytopenia, anemia) and non-hematological (gastrointestinal, genitourinary, skin) toxicities were evaluated.

Clinical and laboratory data were retrieved through electronic medical record review. The following baseline variables were collected the day before starting the treatment: age, gender, disease status, radiotherapy, and chemotherapy regimen administered, hematological and biochemical parameters including white blood cell count (cell/mL), lymphocyte count (cell/mL), neutrophil count (cell/mL), hemoglobin (gr/dl), and platelet count (cell/mL). SII was defined as platelet x neutrophil/lymphocyte.

### 2.2. Statistical Analysis

The primary objective of the present study was to develop a prognostic score based also on inflammatory indexes and to examine its association with DFS and OS in anal cancer patients treated with CT-RT.

Disease-free survival was defined as the time interval between the date of the first cycle of CT-RT to the date of the disease relapse, death, or last follow-up visit. Overall survival was defined as the time interval between the date of the first cycle of CT-RT to the date of death for recurrent disease.

We investigated the correlation between baseline clinical and laboratory variables and DFS and OS using Kaplan–Meier survival curves, and a two-tailed *p*-value < 0.05 was considered statistically significant.

Laboratory variables initially recorded as continuous parameters were later dichotomized relying on the results of Receiver Operating Characteristic (ROC) curves. For SII, hemoglobin, and eosinophils, we used the cut-offs previously identified [8,14,15].

We used Fisher’s exact test or *t* test depending on the nature of the covariates and their characteristics (binary or categorical, respectively). All variables with statistical significance in the univariate analysis were included in a stepwise Cox’s proportional hazard model, and following adjustment for clinical covariates positive at univariate analysis, a multivariate model including our new score system was developed. The decision on which covariate to include in the final index was made taking into consideration their statistical significance, the magnitude of change induced in the analysis, and their clinical plausibility.

A MedCalc package (MedCalc^®^ version 16.8.4) was used for statistical analysis.

## 3. Results

### 3.1. The Hemo-Eosinophil Inflammation Index

A total of 308 consecutive anal cancer patients was available for the present analysis. Patient’s accrual breakdown per institution is as follows: University of Turin, AOU Citta’ della Salute e della Scienza in Turin (94 patients), Department of Modena Cancer Center, Università di Modena e Reggio Emilia (27 patients), San Martino Hospital in Genova (25 patients), Centre Hospitalier Régional Universitaire “Jean Minjoz”, Besançon (93 patients), and European Institute of Oncology, Milan (69 patients).

The main characteristics of the patients enrolled in the study are summarized in Table 1. Adjunctively, 12 patients (4%) were affected with HIV and 104 (34%) presented with any grade lymphopenia.

All covariates were tested within a univariate model. In our sample, HPV status was available only for 116 patients; among these, 107 biopsies resulted in being p16 + ve. Chemotherapy dose modification or reduction was necessary in 30 patients (10%), while radiotherapy treatment breaks ≥3 days were experienced by 20 patients (6%). We devised a prognostic model by combining the inflammation indicators and laboratory parameters investigated in our previous studies (Hb value, SII and eosinophil count) and defined it as the Hemo-Eosinophils Inflammation (HEI) Index. We assessed a weight = 1 to each of the following variables: Hb < 12 g/dL, SII > 560 and eosinophil count ≥100/µL. Accordingly, patients were stratified into two different risk groups as follows: low-risk group (from 0 to 1 negative prognostic factors) and high-risk group (from 2 to 3 negative prognostic factors). For daily clinical practice, we herein provide a link to HEI index calculator (https://casadeigardini.wixsite.com/heiindex (accessed on 14 January 2021)).

### 3.2. Univariate and Multivariate Analysis

Globally, 168 patients were categorized as low-risk and 140 patients were categorized as high risk according to our new scoring system.

Clinical characteristics were well balanced between the low and high-risk group, except for tumor size and global stage, as the number of Stage III patients was significantly higher in the high-risk group (68% vs. 43%, *p* < 0.001) (Table 2). To ensure that this imbalance would not affect our results, we inserted the disease stage in multivariate analysis.

Median DFS for low-risk patients was not reached, while it was found to be 79.5 months [95% Confidence Interval (CI): 45.20–79.54] in the high-risk group. (Figure 1). The difference in DFS was statistically significant between groups (*p* < 0.0001), with an Hazard Ratio (HR) 3.22 (95% CI 2.04–5.10). To evaluate the discriminatory ability of our model, ROC curves were generated. Our model showed a C-index of 0.68 with *p* < 0.001 (Figure 2). The sensitivity was 72% and specificity was 63%. When we analyzed the single components of the HEI index, the area under the ROC curves for Hb < 12, SII > 560 and eosinophil count ≥100 were 0.59, 0.64, and 0.55, respectively. Therefore, the areas under the ROC curves of the single components were lower compared to that pertinent to the HEI Index.

At univariate analysis, excluding the parameters included in the HEI index, nodal status, primary tumor, global stage, and chemotherapy regimen resulted to be the only variables related to DFS with *p* = 0.0021, *p* = 0.0021, *p* = 0.0036, and *p* = 0.0458, respectively (Table 3).

Following adjustment for clinical covariates positive at univariate analysis, multivariate analysis confirmed the HEI index as an independent prognostic factor for DFS (HR: 2.97, 95% CI 1.81–4.90, *p* < 0.0001) (Table 3).

Overall survival curves according to the prognostic model are shown in Figure 1. Median OS for low- and high-risk patients were not reached (Figure 1).

At univariate analysis, excluding the parameters included in the HEI index, nodal status and global stage resulted to be the only variables related to OS with *p* = 0.0309 and *p* = 0.0108, respectively (Table 4).

Following adjustment for clinical covariates positive at univariate analysis, multivariate analysis confirmed HEI index as independent prognostic factor for OS (HR: 2.97, 95% CI 1.36–6.50, *p* = 0.0004) (Table 4).

Considering the high-risk group, 96/140 (68.6%) patients achieved a complete response (CR); on the other hand, in the low-risk group, 147/168 (87.5%) achieved a CR. The odds ratio for complete response in the low-risk patients compared to the group at high-risk according to our prognostic index was 3.20 (1.79–5.73), with *p* = 0.0001. 

### 3.3. Toxicities

Acute gastro-intestinal (GI), genitourinary (GU), dermatologic and hematologic toxicities were assessed during treatment, and these are reported in Table 5.

The toxicity profile in the two groups was equivalent for all the domains considered, except for skin toxicity, which was more likely to be observed in the low-risk group (Table 5).

## 4. Discussion

In this study, we demonstrated that the HEI index, comprising SII, baseline Hb level, and eosinophils levels, is significantly correlated with DFS and OS in anal cancer patients treated with concurrent CT-RT, thus representing a possible tool to predict prognosis in this setting.

We previously demonstrated the prognostic value of high level of SII and baseline eosinophil count, and the level of Hb in this setting [8,14,15]. In the present study, we decided to combine these three parameters in a single score, which is defined as the HEI index, in order to potentially enhance the prognostic value. Interestingly, the C-index resulted for the HEI index was higher than those obtained for the three single parameters, confirming the superiority in terms of prognostic power of our new score. Thus, even though the area under the ROC curve for HEI index did not reach the value of 0.70, which is normally considered to be indicative of good accuracy, we have demonstrated that the combinations of Hb, eosinophils, and SII in a single index could improve the prognostic power.

Even if the survival and local control rates achieved after CT-RT in anal cancer patients is generally high, not all the cases are likely to respond to therapy, and, in the subset of treatment-refractory patients, the prognosis remains dismal. Hence, the use of a prognostic index appears to be attractive, since it could provide a stratification tool to guide the clinical decision-making process. In fact, for patients allocated to the high-risk group, according to the HEI index, intensified treatment strategies may be proposed to improve the outcomes, or reiterative follow-up protocol can be established to detect early recurrence in those achieving a CR after CT-RT.

Since only three prospective trials have previously investigated the role of prognostic factors in this setting [3,7,10], establishing accessible and reproducible prognostic models or nomograms able to stratify patients is of special interest. The HEI index is composed by three data parameters that are easily accessible or deducible from a routine blood examination. Moreover, the three parameters included in the index are related to the process of inflammation–hypoxia–carcinogenesis. In the last decade, the common molecular pathway involving these mechanisms has been deeply investigated, and many inflammation markers were highlighted to predict prognosis and response to cancer treatment [11,12,13,14,15,20,21,22].

Our score includes the SII, which is known to mirror the imbalance between protumor inflammatory pathway activation and antitumor immune function. Previous studies demonstrated that SII in anal cancer patients is correlated with response to CT-RT and, consequently, prognosis [23,24]. We elaborated and validated a nomogram using the SII value, the nodal status, and pre-treatment Hb levels [14]. The SII includes the NLR value, which has been extensively investigated as a biomarker for predicting prognosis in different types of cancer [25,26]. In a previous study published by Toh et al., NLR was highlighted to predict for disease recurrence, OS, and cancer-specific survival in anal cancer patients treated with CT-RT [27]. Moreover, De Felice et al. analyzed a sample of 58 patients in the same setting and concluded that elevated pre-treatment NLR values were correlated with worse prognosis [28]. In a recent analysis performed on 118 anal cancer patients, Knight et al. found the pretreatment NLR not to be associated with treatment outcome nor survival, although the cut-off used was similar to our [29]. The discordance in terms of results could be probably due to the differences in the two sample of patients: in our analysis, the number of patients with nodal involvement was significantly higher compared to those reported in Knight et al., thus leading to the hypothesis that the prognostic value of NLR might be more evident for high-risk patients.

The second feature included in our score is baseline Hb value. In two previous studies, Hb < 12 g/dL was found to be correlated to worse OS and progression-free survival in anal cancer patients treated with CT-RT [8,30]. In their retrospective series, Kapacee et al. settled the Hb cut-off value at 13 g/dL, but the results were in line with the aforementioned reports [31]. Finally, our score included the eosinophils count. Along with lymphocytes, neutrophils, and macrophages, eosinophils are crucial actors in the interplay between inflammation and cancer [32]. A divergent impact on prognosis for eosinophils has been shown according to the type of tumors analyzed: high eosinophil count was associated to a better prognosis in advanced HCC treated with Sorafenib, melanoma, renal carcinoma, colorectal, lung, cervical, and pancreatic cancer, while its prognostic role appeared to be more controversial in breast cancer and lymphoma [33,34,35,36,37,38,39,40,41,42]. In anal cancer, only few data are available on the impact on survival. In a previous work, our group highlighted that low baseline eosinophil count (<100 × 10^9^/L) was correlated to a better DFS (HR = 0.59; *p* = 0.0392) in HPV-positive anal squamous cell cancer [15]. The results of the present study are consistent with our previous study: in particular, we highlighted that high baseline eosinophil count may be associated with a higher risk for recurrence without impacting on survival. Of note, the pathogenesis of anal cancer is notoriously related to high-risk genotypes of Human Papilloma Virus (HPV) infection, which is described in up to 80–90% of cases [43,44]. Since the eosinophils act in both a pro-inflammatory and anti-inflammatory sense depending on the microenvironment and the biological context [45,46], we might suppose that the complex interplay between cancer microenvironment, CT-RT action, and peripheral eosinophils in an HPV-infected organism could result in a drastic immunomodulation and in a weaker RT-antitumor effect, leading to a lower DFS. Our analysis supported the prognostic value of SII, baseline Hb, and eosinophil count, and by combining the three parameters in the HEI index, the prognostic power seems to be improved. It would be interesting to investigate the prognostic value of the HEI index in other types of SCC, first of all Head and Neck (HN) SCCs treated with concurrent CT-RT. In fact, the previous data about the role of inflammation and the well-known involvement of HPV in specific subsites of HN SCCs [47,48] both make this kind of cancer biologically similar to anal SCC, and therefore a good candidate for the investigation of the clinical meaningfulness of the HEI index.

Our work presents some limitations. First, there is the retrospective nature of the study: although we considered different known confounding factors, we could not completely rule out the selection bias (e.g., HPV status). Secondly, the lack of centralization in the evaluation of morphological and functional imaging in assessing disease response may act as an additional confounder and may hamper the statistical power of the DFS evaluation due to inaccuracies. Moreover, we have to note that patients treated in one center (22% of the entire sample) were treated with a different chemotherapy regimen, potentially influencing the prognosis and maybe introducing a selection bias. In fact, it is well known that different chemotherapy regimens as well as different RT doses could have an impact on treatment outcomes and hence generate potential biases [49]. Finally, our score system has to be validated on a large external cohort of patients before being used as a prognostic tool to stratify patients in clinical practice. However, our analysis has been conducted on a large sample size from five different institutions, and the data are consistent with the previous literature. Nevertheless, the low cost and the easy profile make the Hemo-Eosinophils Inflammation index a promising tool for prognostic assessment in anal cancer patients treated with CT-RT.

## 5. Conclusions

In conclusion, we elaborated a new scoring system to assess prognosis (DFS and OS) in anal cancer patients treated with CT-RT with definitive intent. A validation of this new score system is mandatory to translate its use in clinical practice. More investigations are needed in this sense: if confirmed, these observations could add new elements to identify anal cancer patients likely to respond to CT-RT and those at high risk of persistence or relapse to tailor personalized treatment approaches and follow-up strategies [50].

## Figures and Tables

**Figure 1 cancers-13-00671-f001:**
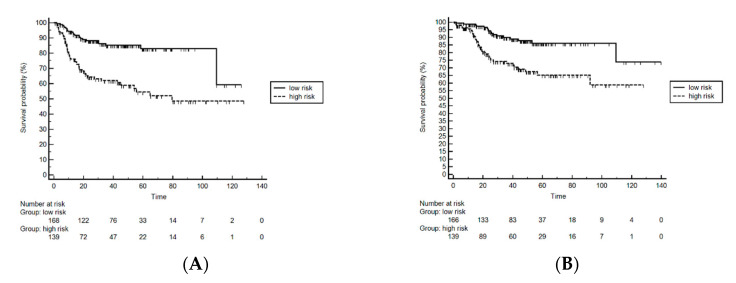
Kaplan–Meier curves for disease-free (**A**) and overall (**B**) survival in high- and low-risk groups according the HEI index.

**Figure 2 cancers-13-00671-f002:**
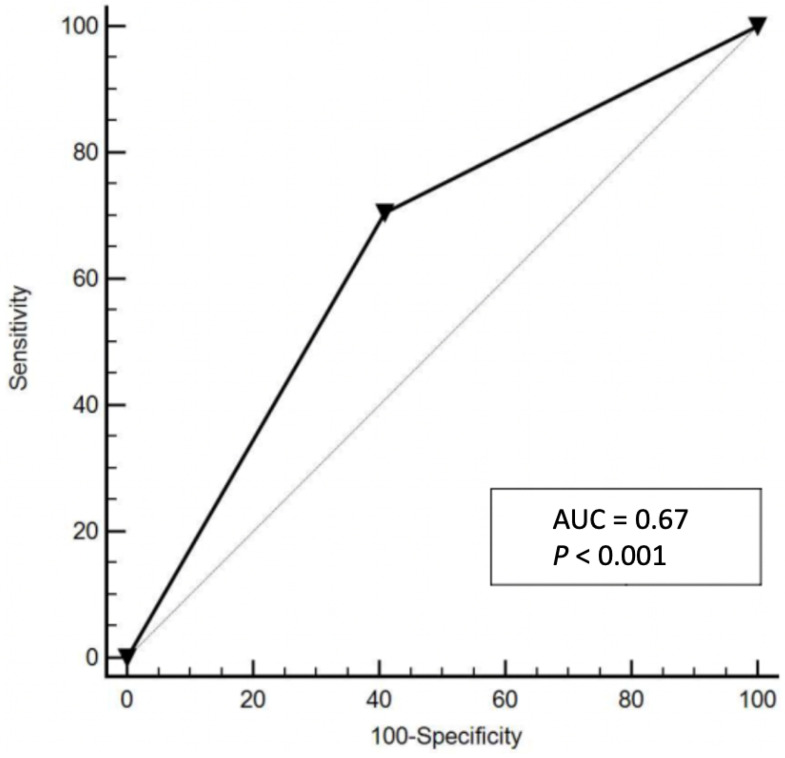
Receiver Operating Characteristic (ROC) curve for HEI index.

**Table 1 cancers-13-00671-t001:** Patient characteristics.

Variable	*N* (%)
Age	-
Median	64
Range	33–92
Gender	-
Female	231 (75.0)
Male	77 (25.0)
T-stage	-
T1–T2	187 (61)
T3–T4	120 (38.5)
Not available	1 (0.5)
N-stage	-
N0	151 (49.0)
N1–N3	155 (50.3)
NA	2 (0.7)
Global Stage	-
I–II	142 (46)
III	165 (53.5)
Not available	1 (0.5)
Grading	-
1–2	164 (53.2)
3	77 (25.3)
Not available	67 (21.5)
Pre-treatment Hb (g/dL)	-
Mean ± SD	13 ± 1.6
Hb < 12 g/dL	251 (81.5)
Hb ≥ 12 g/dL	57 (18.5)
Pre-treatment SII	-
Mean ± SD	779 ± 681
Pre-treatment Eosinophil count (10^3^ cells/µL)	-
Mean ± SD	181 ± 162

**Table 2 cancers-13-00671-t002:** Characteristics in low- and high-risk groups according the Hemo-Eosinophils Inflammation (HEI) index.

Variable	Low Risk *n* (%)	High Risk *n* (%)	*p*
Gender Male Female	- 45 (27) 123 (73)	- 32 (23) 108 (77)	- 0.51
Age <70 ≥70	- 83 (49) 85 (51)	- 62 (44) 73 (56)	- 0.56 -
T-stage T3–T4 T1–T2	- 47 (29) 120 (71)	- 73 (52) 67 (48)	- <0.001
*n*-stage N1–3 N0	- 17 (10) 24 (14)	- 16 (11) 11 (8)	- 0.22
Global Stage I–II III	- 98 (57) 72 (43)	- 46 (32) 95 (68)	<0.001
Grading 1–2 3	- 87 (52) 51 (30)	- 46 (33) 16 (11)	- 0.15

**Table 3 cancers-13-00671-t003:** Univariate and multivariate analysis for disease-free survival.

Variable	Univariate		Multivariate	
-	HR (95% CI)	*p*	HR (95% CI)	*p*
Age (≥70 vs. <70)	0.89 (0.56–1.40)	0.6031	2.25 (1.19–4.26)	0.0120
Gender (Male vs. Female)	1.20 (0.70–2.04)	0.5136	1.19 (1.43–4.72)	0.5711
Chemotherapy (CCDP-based vs. MMC-based)	0.58 (0.33–0.99)	0.0458	0.34 (0.15–0.76)	0.0092
HB (< 12 vs. ≥12 g/dL)	2.30 (1.21–4.37)	0.0102	-	-
SII (>560 vs. ≤560)	1.97 (1.26–3.10)	0.0031	-	-
Eosinophil (≥100 vs. <100/µL)	1.82 (1.07–3.09)	0.0265	-	-
Nodal status (N3 vs. N0–N2)	3.06 (1.50–6.24)	0.0021	-	-
T stage (T4 vs. T1–T3)	3.06 (1.50–6.25)	0.0021	-	-
Stage (III vs. I–II)	1.95 (1.24–3.06)	0.0036	1.39 (0.76–2.54)	0.2836
HEI Index (High Risk vs. Low Risk)	3.22 (2.04–5.10)	<0.0001	2.59 (1.42–4.72)	<0.0001

**Table 4 cancers-13-00671-t004:** Univariate and multivariate analysis for overall survival.

Variable	Univariate		Multivariate	
-	HR (95% CI)	*p*	HR (95% CI)	*p*
Age (≥70 vs. <70)	1.05 (0.61–1.80)	0.8737	1.92 (0.88–4.16)	0.0972
Gender (Male vs. Female)	3.34 (1.75–6.36)	0.0002	1.79 (0.89–3.58)	0.010
Chemotherapy (CCDP-based vs. MMC-based)	0.47 (0.25–0.90)	0.0237	0.25 (0.08–0.79)	0.0186
HB (<12 vs. ≥12 g/dL)	6.68 (3.10–14.41)	<0.0001	-	-
SII (>560 vs. ≤560)	2.11 (1.24–3.60)	0.0062	-	-
Eosinophil (≥100 vs. <100/µL)	1.44 (0.76–2.71)	0.2612	-	-
Nodal status (N3 vs. N0–N2)	1.80 (1.06–3.08)	0.0309	-	-
T status (T4 vs. T1–T3)	2.06 (0.92–4.63)	0.0788	-	-
Stage (III vs. I–II)	2.02 (1.18–3.46)	0.0108	1.97 (0.87–4.42)	0.1018
HEI Index (High Risk vs. Low Risk)	3.01 (1.75–5.17)	0.0001	2.97 (1.36–6.50)	0.0063

**Table 5 cancers-13-00671-t005:** Acute toxicities in low- and high-risk groups.

Toxicity (Any Grade)	Low Risk *n* (%)	High Risk *n* (%)	*p*
Overall Toxicities Yes No	- 158 (94) 10 (6)	- 125 (89) 15 (11)	- 0.15
**Hematological**			
Neutropenia Yes No	- 72 (43) 96 (57)	- 56 (40) 84 (60)	- 0.64
Thrombocytopenia Yes No	- 48 (29) 120 (71)	- 31 (22) 109 (78)	- 0.24
Anemia Yes No	- 41 (24) 127 (76)	- 47 (34) 93 (66)	- 0.1
**Non-Hematological**			
Genitourinary Yes No	- 92 (55) 76 (45)	- 61 (44) 79 (56)	- 0.05
Gastrointestinal Yes No	- 55 (33) 113 (67)	- 44 (31) 96 (69)	- 0.9
Skin Yes No	- 131 (78) 37 (22)	- 88 (63) 52 (37)	- 0.004

## Data Availability

The data presented in this study are available on request from the corresponding author. The data are not publicly available due to privacy reasons.

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
