# Peer review of "The Prognostic Value of the New Combined Hemo-Eosinophil Inflammation Index (HEI Index): A Multicenter Analysis of Anal Cancer Patients Treated with Concurrent Chemo-Radiation"

_cancers, 2021, doi:10.3390/cancers13040671_

Round 1

Reviewer 1 Report

This is an original investigation on a new score system to evaluate prognosis in patients with anal cancer submitted to chemo-radiotherapy.

ROC curve analysis was performed, AUC for HEI index was 0.68 and AUCs for the single components of the index were lower (0.59, 0.64 and 0.55).  Sensibility and specificity for this new index should be reported. An AUC<0.70 indicates poor accuracy, the Authors should comment on that in the discussion.

The chemotherapy regimen in one of the 5 institutions was different and this may have well influenced patients’ prognosis, this issue should be discussed.

Percentage of stage III patients is not reported correctly in section 3.2 (line 158).

There is a very recent report (Knight K, Clin Oncol 2021) evaluating the role of systemic inflammation as prognostic indicators in anal cancer. This article should be included in the discussion of NLR value as biomarker.

Author Response

This is an original investigation on a new score system to evaluate prognosis in patients with anal cancer submitted to chemo-radiotherapy.

Thank you.

ROC curve analysis was performed, AUC for HEI index was 0.68 and AUCs for the single components of the index were lower (0.59, 0.64 and 0.55).  Sensibility and specificity for this new index should be reported. An AUC<0.70 indicates poor accuracy, the Authors should comment on that in the discussion.

REPLY: We thank very much the reviewer for this point. We’ve reported an observation more about the C index and the AUC for the HEI compared the ones for the single components of the index. Only for the reviewer, the sensitivity was 72% and specificity was 63%.

The chemotherapy regimen in one of the 5 institutions was different and this may have well influenced patients’ prognosis, this issue should be discussed.

REPLY: We do thank the reviewer for this point. By performing both univariate and multivariate analysis, we highlighted that the choice of a CT regimen vs  the other had a significant impact on  DFS and OS. Therefore, we added these results in Tables 3 and 4. Unfortunately, since only a single institution treated patients with cisplatin and 5-FU, it is difficult to evaluate if patients really benefited more from this CT regimen (Cis and 5FU) compared the standard (mytomicin and 5FU). In fact, due to the retrospective nature of the trial, and even if we considered different known confounding factors, we could not completely rule out the selection bias.

The aim of the present study was to build a new score system, but surely these results suggested us a new point to focus on in a further paper, eventually by performing a propensity score matched analysis.

Percentage of stage III patients is not reported correctly in section 3.2 (line 158).

REPLY: We apologize with the reviewer for this point. We corrected it.

There is a very recent report (Knight K, Clin Oncol 2021) evaluating the role of systemic inflammation as prognostic indicators in anal cancer. This article should be included in the discussion of NLR value as biomarker.

REPLY: Dear Reviewer, thank You for your valuable suggestion. We added a comment about the discordance in terms of result about the NLR prognostic value between our analyses and those of the colleagues.

Reviewer 2 Report

2 Materials and Methods

2.1 Patient selection

Please provide confirmation numbers of the respectives ethics review board (at least for the leading institution)

Please provide inclusion period (start year - end year) of patients

Please provide details of number of patients included into retrospecitve analysis per institution

If possible, please provide information about imaging for staging: CT-based oder PET-based and number of patients for each technique

Please provide details about percentages of patients treated with 5FU/MMC and 5FU/CDDP or Cap/CDDP, respectively

Please provide details about radiotherapy treatment:

a) Percentages of patients treated to 50.4 Gy and 59.4 Gy, respectively, also provide information how many patients received inguinal radiotherapy

b) PTV volume: average, min, max, stdv

c) Field borders cranially and caudally

d) if possible:percentage of pelvic bone structure/bone marrow in the treatment fields

e) SIB Technique: if yes dose per fraction per dose level

Please provide information about hematological status of patients prior to treatment (so it can be compared to hematological toxicities during/after treatment

a) percentage of patients with anemia (Hb under 12). According to Table 1 hardly any patient was anemic prior to treatment. I am asking because according to my understanding building your new risk model Hb<12 g/l was a factor, and then of not much significance in the model as anemia was not an issue.

b) percentage of patients with immune deficiency (eg. HIV) and lymphopenia

c) percentage of patients receiving blood transfusions before or during treatment

Please provide information on how many paitents could not receive chemotherapy as planned (eg. dose reduciton, only one cycle etc)

Please provide information on how many patients did not complete radiotherapy as planned

Please provide percentage of patients HPV/p16 positive. Any chance to perform UVA or MVA with respect to HPV?

As stated correctly HEI index has to be validated independently on another data set, preferrably on prospective data or even in a clinical trial, before this index should be used in daily practice.

It would be worthwile to mention pontential biases eg. role of chemotherapy, RT dose and field size as this influences the hematopoetic system

Did patients receive GM-CSF during chemotherapy?

Do you expect that the SII Index (defined as platelet x neutrophil/lymphocyte) is influenced by stystemic therapy (eg. drop in platelets and neutrophils/lympocytes?) How would this affect your propsed HEI Index?

As the SII index is already a composite index, how do you ensure not introducing a bias in the composite index HEI?

3 Results

Please explain why 75% of study popultion are female. Is this by chance or due to spefiic study recruitment procedures (Table 1)

What do you mean by Globar state, is this equal to the AJCC stage system? Please specify (Table 1)

Please explain why in 21.5% of cases grading was not documented (Table 1)

3.1 HEI index

Please explain in more detail the scientific rational to create the HEI index. Did did you have a hypothesis why exactely SII, baseline hemoglobin an eosinophils and not other factors contribute to this new and proposed index? Did you check if with only two factors, eg. SII and hemoglobin, DFS was also favorable? This should also be discussed in more detail in the section discussion

SII is defined as platelet x neutrophil/lymphocyte

Fig 2: Please show other ROCs for HB, SII and Eos

Page 6/11: it is stated for MVA: HEI index for DFS 2.88, 95% CI 1.73-4.80. These numbers are different to what is documented for HEI Index in Table 3

What could be the reason for higher skin toxicity in the low risk group? This is unexpected.

4 Discussion

The discussion is well written and interesting to read. However, description of a hypothesis why HEI index comprising the three values SII, Hb and Eos should be further investigated, in particular what is the biological mechanism and would it work in other SCC entities, eg. H&N cancer? A broader view where the HEI index could be used would enrich the discussion.

Author Response

2.1 Patient selection

Please provide confirmation numbers of the respectives ethics review board (at least for the leading institution)

REPLY: Dear Reviewer, thank You for Your observation. We added the information required.

Please provide inclusion period (start year - end year) of patients

REPLY: We provided the period of treatment.

Please provide details of number of patients included into retrospective analysis per institution

REPLY: We provided details of number of patients per institution.

If possible, please provide information about imaging for staging: CT-based oder PET-based and number of patients for each technique

REPLY: We added the information required in the section 2.1

Please provide details about percentages of patients treated with 5FU/MMC and 5FU/CDDP or Cap/CDDP, respectively

REPLY: We added the information required in the section 2.1

Please provide details about radiotherapy treatment:

  1. a) Percentages of patients treated to 50.4 Gy and 59.4 Gy, respectively, also provide information how many patients received inguinal radiotherapy

Radiotherapy was delivered using static or volumetric intensity modulated approaches up to a total prescription dose to the macroscopic disease of 50.4 Gy (48 patients; 15%), 54 Gy (215 patients; 70%) or 59.4 Gy (45 patients; 15%) in 28, 30 or 33 fractions respectively, depending on tumor size and stage. Elective nodal irradiation was offered to patients on pelvic lymph-nodes and inguinal groins up to a conventionally fractionated dose of 45 Gy in 30 fractions. The inguinal region was comprised within elective treatment volumes in 285 patients (92%).

  1. b) PTV volume: average, min, max, stdv

We apologize, but we are not able to retrieve data on the cm3 of PTV volumes.

  1. c) Field borders cranially and caudally

The caudal limit of treatment volumes is at the anal verge or areas of peri-anal skin involvement. The most cephalad aspect is where the common iliac vessels bifurcate into external/internal iliacs (approximate boney landmark: sacral promontory).

  1. d) if possible:percentage of pelvic bone structure/bone marrow in the treatment fields

The mean relative overlap volume was 12.2% (SD: ± 5.2%)

  1. e) SIB Technique: if yes dose per fraction per dose level

For patients receiving a simultaneous integrated boost (SIB), 1.8 Gy were given daily to the macroscopic disease and 1.55 Gy were daily administered to the elective volumes.

Please provide information about hematological status of patients prior to treatment (so it can be compared to hematological toxicities during/after treatment

  1. percentage of patients with anemia (Hb under 12). According to Table 1 hardly any patient was anemic prior to treatment. I am asking because according to my understanding building your new risk model Hb<12 g/l was a factor, and then of not much significance in the model as anemia was not an issue.

REPLY: Thank You for the observation. We added in Table 1 the number of patients with Hb ≥ and < 12 g/dl thus making the concept clearer. The data previously reported in the Table (13 g/dl) was the median of hemoglobin values in the sample of patients. In our population the majority of patients were observed having a lower hemoglobin (81.5%).

  1. percentage of patients with immune deficiency (eg. HIV) and lymphopenia

REPLY: 12 patients (4%) were affected with HIV and 104 patients started treatment with lymphopenia at baseline. We added these data in the text.  

  1. percentage of patients receiving blood transfusions before or during treatment

REPLY: This is a very interesting point. In our cohort no patients received blood transfusions before and during treatment.

Please provide information on how many patients could not receive chemotherapy as planned (eg. dose reduction, only one cycle etc). Please provide information on how many patients did not complete radiotherapy as planned

REPLY: We added this point in the section 3.1. Chemotherapy dose modification or reduction was necessary in 30 patients (10%), while radiotherapy treatment breaks > 3 days were experience by 20 patients (6%)

Please provide percentage of patients HPV/p16 positive. Any chance to perform UVA or MVA with respect to HPV?

REPLY: HPV status was available only for 116 patients; of these 107 biopsies resulted in being p16 +ve.  Hence, we could not include this factor in univariate nor multivariate analyses. We added this point to the limitations of our study.

As stated correctly HEI index has to be validated independently on another data set, preferrably on prospective data or even in a clinical trial, before this index should be used in daily practice.

REPLY: We thank very much the reviewer for this point; We completely agree with her/him and we will start with a prospective study for validate our index; In the conclusion we had put “A validation of this new score system is mandatory to translate its use in clinical practice.”

It would be worthwile to mention pontential biases eg. role of chemotherapy, RT dose and field size as this influences the hematopoetic system

REPLY: We thank the reviewer for having raised this point; we completely agree with her/him. We have specified that in the line 302-306.

Did patients receive GM-CSF during chemotherapy?

REPLY: Unfortunately, the retrospectivity nature of this study could not allow us to collect data about the use of GM-CSF during the treatment.

Do you expect that the SII Index (defined as platelet x neutrophil/lymphocyte) is influenced by stystemic therapy (eg. drop in platelets and neutrophils/lympocytes?) How would this affect your propsed HEI Index?

REPLY: This is a very interesting point and we thank the reviewer for this point. Of course, SII index change during the treatment due to the treatment but HEI index was built on the data collected before starting the treatment and not during the treatment. It will be very interesting evaluate the change of HEI index during the treatment and the clinical outcome in a future study. Thank you, reviewer, for this suggestion.

As the SII index is already a composite index, how do you ensure not introducing a bias in the composite index HEI?

We provided a measure associated to the uncertainty of the prognostic capacity of HEI, by providing sensitivity and specificity of the HEI index to outline its discriminatory power.

3 Results

Please explain why 75% of study population are female. Is this by chance or due to spefiic study recruitment procedures (Table 1)

REPLY: We didn’t select the patients on the basis of gender. As also reported by American cancer society (https://www.cancer.org/cancer/anal-cancer/about/what-is-key-statistics.html) the majority of cases with anal cancer is female. In US (but it is the same in Europe) about 9,090 new cases are expected at year, of these 6,070 are women (66.8%) and are 3,020 men.

Please explain why in 21.5% of cases grading was not documented (Table 1)

REPLY: Dear Reviewer, unfortunately by performing a tissue biopsy in many cases the grade could not be defined: this is the reason why we miss a significant percentage of data about grade.

3.1 HEI index

Please explain in more detail the scientific rational to create the HEI index. Did did you have a hypothesis why exactely SII, baseline hemoglobin an eosinophils and not other factors contribute to this new and proposed index? Did you check if with only two factors, eg. SII and hemoglobin, DFS was also favorable? This should also be discussed in more detail in the section discussion

REPLY: the rationale of building a score including the three parameters hemoglobin, eosinophils and SII was settled on our previous works. Starting from the growing evidences about the role of inflammation in cancer, we previously demonstrated the prognostic value of hemoglobin, SII and eosinophils in the same setting (anal SCC patients treated with concurrent CT-RT). For this reason, we thought to unify these three parameters in a single score in order to potentiate the prognostic value. Moreover, by performing a ROC analysis, we demonstrated that effectively the C-index of the HEI index was superior to those of the single components. We explained the rationale in the text in the lines 223-228.

SII is defined as platelet x neutrophil/lymphocyte

REPLY: We thank the reviewer for this point and we apologies to her/him for this. We added the definition in the text.

Fig 2: Please show other ROCs for HB, SII and Eos

REPLY: Only for the reviewer we reported here the ROC curves. It is interesting to note that in particular for eosinophil and Hb the curves were in favor to specificity (eosinophil) or sensitivity (for Hb).

Page 6/11: it is stated for MVA: HEI index for DFS 2.88, 95% CI 1.73-4.80. These numbers are different to what is documented for HEI Index in Table 3

REPLY: We very thank the reviewer for this point and we apologies to her/him for this. We have adjusted the text.

What could be the reason for higher skin toxicity in the low risk group? This is unexpected.

REPLY: Dear Reviewer, correctly it was an unexpected finding. This could probably due to an unbalance in risk factors or characteristics favoring toxicity between the 2 groups, that we could not account for nor double-check for due to the retrospective nature of our study.

4 Discussion

The discussion is well written and interesting to read. However, description of a hypothesis why HEI index comprising the three values SII, Hb and Eos should be further investigated, in particular what is the biological mechanism and would it work in other SCC entities, eg. H&N cancer? A broader view where the HEI index could be used would enrich the discussion.

REPLY: We thank the reviewer for her/his comment. Since we already published several works about this topic, we started from our previous results to build an index able to potentiate the prognostic value and, therefore, to better stratify patients.
